# Three-Dimensional Image Visualization under Photon-Starved Conditions Using *N* Observations and Statistical Estimation

**DOI:** 10.3390/s24061731

**Published:** 2024-03-07

**Authors:** Hyun-Woo Kim, Min-Chul Lee, Myungjin Cho

**Affiliations:** 1Department of Computer Science and Networks, Kyushu Institute of Technology, 680-4 Kawazu, Iizuka-shi 820-8502, Fukuoka, Japan; kim@ois3d.cse.kyutech.ac.jp (H.-W.K.); lee@csn.kyutech.ac.jp (M.-C.L.); 2School of ICT, Robotics, and Mechanical Engineering, Hankyong National University, IITC, 327 Chungang-ro, Anseong 17579, Kyonggi-do, Republic of Korea

**Keywords:** maximum likelihood estimation, N observations, photon-counting integral imaging, three-dimensional imaging, volumetric computational reconstruction

## Abstract

In this paper, we propose a method for the three-dimensional (3D) image visualization of objects under photon-starved conditions using multiple observations and statistical estimation. To visualize 3D objects under these conditions, photon counting integral imaging was used, which can extract photons from 3D objects using the Poisson random process. However, this process may not reconstruct 3D images under severely photon-starved conditions due to a lack of photons. Therefore, to solve this problem, in this paper, we propose *N*-observation photon-counting integral imaging with statistical estimation. Since photons are extracted randomly using the Poisson distribution, increasing the samples of photons can improve the accuracy of photon extraction. In addition, by using a statistical estimation method, such as maximum likelihood estimation, 3D images can be reconstructed. To prove our proposed method, we implemented the optical experiment and calculated its performance metrics, which included the peak signal-to-noise ratio (PSNR), structural similarity (SSIM), peak-to-correlation energy (PCE), and the peak sidelobe ratio (PSR).

## 1. Introduction

Three-dimensional (3D) image visualization under photon-starved conditions is challenging nowadays. There are several 3D imaging techniques, such as light detection and ranging (LiDAR), stereoscopic imaging, holography, and integral imaging. LiDAR [1] can detect the shape of 3D objects by detecting and measuring the time it takes for light to be reflected from the objects; however, it is not cost-effective and it may not record the image’s information (i.e., the RGB map). Stereoscopic imaging [2,3,4,5,6] can obtain 3D information by analyzing the disparity of stereoscopic images (i.e., binocular images), and it is simple to implement because it uses only several cameras; however, it may not provide high resolution 3D information and full parallax due to the limit of the unidirectional disparity of stereoscopic images. On the other hand, holography [7,8,9,10,11] can provide full parallax and continuous viewing points of 3D images by using coherent light sources. The term is derived from a Greek word meaning “perfect imaging”. However, it is very complicated to implement because its system alignment is very sensitive and it uses a lot of optical components, such as a beam splitter, a laser, and a spatial light filter.

To overcome the problems in the 3D imaging techniques mentioned above, integral imaging, which was first proposed by G. Lippmann [12], has been studied. Integral imaging can provide full parallax and continuous viewing points without any special viewing devices and coherent light sources. However, it has several drawbacks such as low viewing resolution, a narrow viewing angle, and a shallow depth of focus. Several studies have been conducted to solve these disadvantages [13,14,15,16,17,18,19,20,21,22,23,24,25,26,27]. In addition, 3D imaging techniques may not provide 3D information of objects under photon-starved conditions due to the lack of photons. To obtain 3D information of objects under these conditions, photon-counting integral imaging has been studied [28,29,30,31,32,33,34,35,36,37].

Photon-counting integral imaging, which uses photon-counting imaging and integral imaging, can record and reconstruct 3D images under photon-starved conditions. Photon-counting imaging can detect photons that rarely occur in unit time and space under photon-starved conditions by using the computational photon-counting model [38]. The computational photon-counting model follows a Poisson distribution due to the characteristics of photon occurrence. Therefore, it can record the images randomly from the original scene with the expected number of photons. However, it may not obtain the images when the light intensity is very low (i.e., the light intensity is close to zero) since the Poisson distribution will return zero values. In addition, to estimate the images of the original scene statistically under photon-starved conditions, many samples are required; therefore, photon-counting integral imaging has been proposed [28,29,30,31,32,33,34]. Photon-counting integral imaging uses multiple 2D images with different perspectives from 3D objects through a lenslet or camera array, where these images are referred to as elemental images. Three-dimensional information of a scene can be estimated using statistical estimation techniques such as maximum likelihood estimation or Bayesian estimation [29]. However, it still exhibits low visual quality due to the randomness of the Poisson distribution and the lack of photons.

In this paper, we propose a method for 3D photon-counting integral imaging with multiple observations to enhance the visual quality of photon-counting images and reconstructed 3D images under photon-starved conditions. From the original scene, multiple photon-counting images, which are generated by photon-counting imaging, are statistically independent. Thus, randomness in the Poisson distribution can be overcome. In addition, the accuracy of statistical estimation can be improved due to the increase in the number samples. Therefore, our proposed method improves the visual quality of photon-counting images and reconstructed 3D images under photon-starved conditions by photon-counting integral imaging with *N* observations. To verify the visual quality enhancement of our proposed method, experimental results are shown and numerical results such as the peak signal-to-noise ratio (PSNR), structural similarity (SSIM), peak-to-correlation energy (PCE), and the peak sidelobe ratio (PSR) are calculated.

This paper is organized as follows. We present the basic concept of photon-counting integral imaging and *N*-observation photon-counting integral imaging in Section 2. We give the experimental results along with a discussion of these results in Section 3. Finally, we conclude our work with a summary in Section 4.

## 2. *N*-Observation Photon-Counting Integral Imaging

In this section, we describe the basic concept of integral imaging, photon-counting imaging, photon-counting integral imaging, and *N*-observation photon-counting integral imaging. Photon-counting imaging is used to record an image under photon-starved conditions. To visualize a 3D image under these conditions, photon-counting integral imaging uses statistical estimations such as maximum likelihood estimation (MLE) and Bayesian approaches with a uniform distribution and a gamma distribution as prior information, respectively. However, under severely photon-starved conditions, photon-counting integral imaging may not visualize 3D objects for object recognition. To overcome this problem, in this paper, we propose photon-counting integral imaging with *N* observations and verify the visual quality enhancement of our proposed method by optical experiments and numerical results.

### 2.1. Integral Imaging

Integral imaging, which was first proposed by G. Lippmann in 1908, can provide full parallax and continuous viewing points of color 3D images without any special viewing glasses. In addition, it does not require a coherent light source such as a laser. It consists of two processes: pickup and display (or reconstruction). Figure 1 illustrates the basic concept of integral imaging. In the pickup stage, as shown in Figure 1a, rays coming from the 3D object are recorded through a lens array. These rays generate multiple 2D images with different perspectives for the 3D object, which are referred to as elemental images. Then, in the display stage, as depicted in Figure 1b, elemental images are projected through the homogeneous lens array in the pickup stage. Finally, a 3D image can be observed by the human eye without any special viewing glasses. In addition, this 3D image has full parallax and continuous viewing points. However, as shown in Figure 1b, the depth of the 3D image is reversed, and is referred to as the pseudoscopic real image. To solve this problem, Arai et al. [14] proposed a pseudoscopic to orthoscopic conversion (i.e., a P/O conversion) by rotating each elemental image through 180°. Thus, using a P/O conversion, the orthoscopic virtual 3D image can be displayed in space. However, the resolution of each elemental image is very low since the resolution of the image sensor is divided by the number of lenses. Therefore, the resolution of the 3D image in the display stage may be insufficient for observers.

To solve this resolution problem, synthetic aperture integral imaging (SAII) was proposed by Jang et al. [16]. Figure 2 shows the schematic of SAII, which uses a camera array instead of a lens array to obtain a high resolution for each elemental image. In lens-array-based integral imaging, the pitch between lenses is stationary while, in SAII, the pitch between cameras may be dynamic. Therefore, it is possible to obtain higher 3D resolution (i.e., lateral and longitudinal resolutions) than when using lens-array-based integral imaging. However, since the resolution of each elemental image is high, the conventional display panel may not display elemental images in the 3D image display. Thus, volumetric computational reconstruction (VCR) was proposed [19,27].

VCR can provide a sectional image at the reconstruction depth for a 3D image using high resolution elemental images produced by SAII. It projects elemental images through a virtual pinhole array on the reconstruction plane where each elemental image is shifted, as illustrated in Figure 3. VCR can be implemented as follows [19,27]:(1)$$\Delta x=\left\lfloor \frac{N_{x}fp_{x}}{c_{x}z_{r}}\right\rceil ,\;\Delta y=\left\lfloor \frac{N_{y}fp_{y}}{c_{y}z_{r}}\right\rceil$$
where $N_{x},N_{y}$ are the number of pixels for each elemental image; *f* is the focal length of the virtual pinhole; $p_{x},p_{y}$ are the pitches between the virtual pinholes; $c_{x},c_{y}$ are the sensor sizes; and $z_{r}$ is the reconstruction depth. Finally, various sectional images at different reconstruction depths for a 3D image can be obtained as follows:(2)$$I\left(x,y,z_{r}\right)=\frac{1}{O(x,y,z_{r})}\sum_{k=1}^{K}\sum_{l=1}^{L}E_{kl}\left(x+(k-1)\Delta x,y+(l-1)\Delta y\right)$$
where $E_{kl}$ is the *k*th column and *l*th row elemental image produced using SAII and $O(x,y,z_{r})$ is the overlapping matrix at the reconstruction depth, $z_{r}$.

In the VCR mentioned above, since the number of shifting pixels is stationary for each elemental image, the quantization error may increase. For example, when Δx=2.3 and the number of elemental images in the *x* direction is 10, and the shifting pixel values are calculated using Equation (Equation 1): Δx=2 and $\Delta x_{0}=0,\;\Delta x_{1}=2,\;\Delta x_{2}=4,\ldots$. Thus, 0, 0.3, 0.6, … quantization errors may occur for each shifting pixel value. These quantization errors may cause degradation in the visual quality of the 3D images. Therefore, we need to reduce this quantization error for VCR. In this paper, we utilize VCR with nonuniform shifting pixels. In this technique, the shifting pixel values are calculated as follows [27]:(3)$$\Delta x_{i}=\left\lfloor \frac{N_{x}fp_{x}}{c_{x}z_{r}}\times \left(i-1\right)\right\rceil ,\Delta y_{j}=\left\lfloor \frac{N_{y}fp_{y}}{c_{y}z_{r}}\times \left(j-1\right)\right\rceil$$
By Equation (Equation 3), the shifting pixel values are calculated: $\Delta x_{0}=0,\;\Delta x_{1}=2,\;\Delta x_{2}=5,\ldots$. Thus, 0, 0.3, 0.4, … quantization errors for each shifting pixel value occur. Note that a VCR with nonuniform shifting pixels has lower quantization error than a VCR with stationary shifting pixels. Therefore, the depth resolution of the 3D images may be enhanced in VCR with nonuniform shifting pixels. In this paper, we use this VCR method for reconstructing 3D images expressed by [27].
(4)$$I\left(x,y,z_{r}\right)=\frac{1}{O(x,y,z_{r})}\sum_{k=1}^{K}\sum_{l=1}^{L}E_{kl}\left(x+\Delta x_{k},y+\Delta y_{l}\right)$$

### 2.2. Photon-Counting Imaging

Under photon-starved conditions, the number of photons reflected from objects may be low. Thus, it may be difficult to record the images of objects under these conditions. To solve this problem, photon-counting imaging has been proposed. Photons emitted from objects can be recorded by a physical photon-counting detector, as shown in Figure 4. However, it is difficult to adjust the number of extracted photons that are emitted from objects. In addition, it is very expensive to implement. Therefore, in this paper, we introduce a computational photon-counting model that can be realised using a Poisson distribution since photons may occur rarely in unit time and space [38]. Figure 5 describes the computational photon-counting model. First, the original image, I(x,y), is normalized so that the image has unit energy to control the number of photons. Then, multiplying the number of extracted photons, $N_{p}$, by the normalized image, λ(x,y), and implementing a Poisson random process, aphoton-counting image, C(x,y), can be obtained. The computational photon-counting process is expressed as follows [28,29]:(5)$$\lambda \left(x,y\right)=\frac{I(x,y)}{\sum_{x}\sum_{y}I\left(x,y\right)}$$
(6)$$C\left(x,y\right)|N_{p}\lambda \left(x,y\right)∼Poisson\left(N_{p}\lambda \left(x,y\right)\right)$$
where $N_{p}$ is adjustable. When $N_{p}$ is large more photons can be extracted and vice versa.

If an object or scene has singular pixel intensity (i.e., an extremely high pixel value compared with others) most photons are extracted from singular pixels, as shown in Figure 6. Thus, image visualization or recording may be difficult. To overcome this difficulty, photon-counting imaging can be applied after choosing a region of interest (ROI) that does not contain any singular pixels. In addition, the visual quality of a photon-counting image may be degraded due to photon-starved conditions, as shown in Figure 7. To solve these problems, a more accurate estimation method with a lot of samples (i.e., more photons) is required. Since integral imaging can produce multiple 2D images with different perspectives via a lens array or a camera array, it can provide a solution by using a statistical estimation method, such as maximum likelihood estimation (MLE) and Bayesian approaches. Photon-counting integral imaging is presented in next subsection.

### 2.3. Photon-Counting Integral Imaging

Integral imaging can record multiple 2D images with different perspectives from 3D objects through a lens array or a camera array. Here, these 2D images are referred to as elemental images and they can be sampled to estimate a scene of objects in photon-counting imaging with statistical estimation. Therefore, photon-counting integral imaging can be utilized to reconstruct an image under photon-starved conditions. To describe photon-counting integral imaging, we need to consider a statistical estimation method, such as MLE.

Photon-counting images generated from elemental images by integral imaging are followed by the application of a Poisson distribution. Thus, the likelihood function can be written as follows [28,29]:(7)$$P\left(C_{kl}|N_{p}\lambda_{kl}\right)=\frac{{\left(N_{p}\lambda_{kl}\right)}^{C_{kl}}e^{-N_{p}\lambda_{kl}}}{C_{kl}!}$$
(8)$$L\left(N_{p}\lambda_{kl}|C_{kl}\right)=\prod_{k=1}^{K}\prod_{l=1}^{L}\frac{{\left(N_{p}\lambda_{kl}\right)}^{C_{kl}}e^{-N_{p}\lambda_{kl}}}{C_{kl}!}$$
where $\lambda_{kl}$ is the normalized irradiance of the elemental image in the *k*th column and the *l*th row calculated using Equation (Equation 5), $C_{kl}$ is its photon-counting image calculated using Equation (Equation 6), and K,L are the number of photon-counting images in the *x* and *y* directions, respectively.

By taking the logarithm and maximizing Equation (Equation 8), the estimated scene can be obtained as follows [28,29]:(9)$$\log\left[L\left(N_{p}\lambda_{kl}|C_{kl}\right)\right]=l\left(N_{p}\lambda_{kl}|C_{kl}\right)\propto \sum_{k=1}^{K}\sum_{l=1}^{L}C_{kl}\log N_{p}\lambda_{kl}-\sum_{k=1}^{K}\sum_{l=1}^{L}N_{p}\lambda_{kl}$$
(10)$$\frac{\partial l(N_{p}\lambda_{kl}|C_{kl})}{\partial \lambda_{kl}}=\frac{C_{kl}}{\lambda_{kl}}-N_{p}=0$$
(11)$${\hat{\lambda }}_{kl}=\frac{C_{kl}}{N_{p}}$$

Finally, using volumetric computational reconstruction (VCR) with the nonuniform shifting pixel of integral imaging, 3D reconstructed images under photon-starved conditions are obtained by [28,29]:(12)$$\hat{I}\left(x,y,z_{r}\right)=\frac{1}{O(x,y,z_{r})}\sum_{k=1}^{K}\sum_{l=1}^{L}{\hat{\lambda }}_{kl}\left(x+\Delta x_{k},y+\Delta y_{l}\right)$$
where $O(x,y,z_{r})$ is the overlapping matrix at the reconstruction depth, $z_{r}$. Figure 8 shows the results under photon-starved conditions using Equations (Equation 7)–(Equation 12). As shown in Figure 8, the reconstructed 3D images produced by photon-counting integral imaging with MLE (see Figure 8b,c) have better visual quality than the photon-counting image (see Figure 8a).

Since severely photon-starved conditions have only a few photons, photon-counting integral imaging may not reconstruct 3D images. This implies that an accurate scene with enhanced visual quality may not be visualized under these conditions; therefore, to solve this problem, in this paper, we propose photon-counting integral imaging with *N* observations.

### 2.4. N-Observation Photon-Counting Integral Imaging

In photon-counting imaging, photons may be detected randomly with probabilities from a Poisson distribution. Photon-counting images with single observation by Equations (Equation 5) and (Equation 6) may not be visualized well under severely photon-starved conditions as only a few photons are detected from the scene. This means that the number of photons (i.e., the number of samples) can determine the visual quality of a photon-counting image under these conditions. To increase the number of photons, in this paper, we observe *N* photon-counting images from a scene. Then, using a statistical estimation method, such as MLE, we estimate the scene under severely photon-starved conditions. Here, this method is called *N*-observation photon-counting imaging.

In *N*-observation photon-counting imaging, multiple photon-counting images can be generated randomly through a Poisson distribution as follows:(13)$$C_{n}\left(x,y\right)|N_{p}\lambda \left(x,y\right)∼Poisson\left(N_{p}\lambda \left(x,y\right)\right),\;n=1,2,\ldots ,N$$
where *n* is the index of the *N*th observation for a photon-counting image. Now, *N* photon-counting images can construct the likelihood function for each elemental image since they are statistically independent.
(14)$$P\left(C_{n}|N_{p}\lambda \right)=\frac{{\left(N_{p}\lambda \right)}^{C_{n}}e^{-N_{p}\lambda }}{C_{n}!}$$
(15)$$L\left(N_{p}\lambda |C_{n}\right)=\prod_{n=1}^{N}\frac{{\left(N_{p}\lambda \right)}^{C_{n}}e^{-N_{p}\lambda }}{C_{n}!}$$
For convenience of calculation, we take the logarithm of the likelihood function as follows:(16)$$\log\left[L\left(N_{p}\lambda |C_{n}\right)\right]=l\left(N_{p}\lambda |C_{n}\right)\propto \sum_{n=1}^{N}C_{n}\log N_{p}\lambda -\sum_{n=1}^{N}N_{p}\lambda$$
Then, using MLE, a scene (i.e., elemental image) under photon-starved conditions can be estimated by the following equations.
(17)$$\frac{\partial l(N_{p}\lambda |C_{n})}{\partial \lambda }=\frac{\sum_{n=1}^{N}C_{n}}{\lambda }-NN_{p}=0$$
(18)$$\tilde{\lambda }\left(x,y\right)=\frac{\sum_{n=1}^{N}C_{n}\left(x,y\right)}{NN_{p}}$$

Now, a scene under photon-starved conditions is estimated by *N*-observation photon-counting imaging: the estimated scene is the average of the *N* observed photon-counting images. Thus, the visual quality of a photon-counting image may be enhanced statistically. For a 3D reconstruction of a scene under these conditions, VCR with a nonuniform shifting pixel of integral imaging is utilized as follows:(19)$$\tilde{I}\left(x,y,z_{r}\right)=\frac{1}{O(x,y,z_{r})}\sum_{k=1}^{K}\sum_{l=1}^{L}{\tilde{\lambda }}_{kl}\left(x+\Delta x_{k},y+\Delta y_{l}\right)$$

Figure 9 shows the results obtained by conventional photon-counting integral imaging and our proposed method with 100 observations under severely photon-starved conditions ($N_{p}$ = 5000). As shown in Figure 9, the reconstructed 3D image produced by our proposed method has better visual quality than the reconstructed 3D image produced by the conventional method, which has more noise.

## 3. Experimental Results

### 3.1. Experimental Setup

To show the feasibility of our proposed method, we implemented the optical experiment. Figure 10 illustrates the experimental setup. To capture the elemental images with high resolution, we utilized SAII with a camera (Nikon D850, Tokyo, Japan) array and a white light. The focal length of the camera, *f*, was 48 mm; the pitch between cameras, *p* was 2 mm; the first object (red car) was located 190 mm from the camera array; and the second object (white car) was located 230 mm from the camera array. The resolution of each elemental image was 752(H) × 500(V), and the number of elemental images was 10(H) × 10(V). To observe the reconstructed 3D images with the human eye, we set the expected number of photons as $N_{p}$ = 18,800, which is 0.05 photons per pixel of each elemental image. In addition, the range of observations was set from 1 to 100.

### 3.2. Experimental Results

To generate photon-counting images, Equations (Equation 5) and (Equation 6) were applied to elemental images by SAII. Figure 11 shows the photon-counting images with different observations, where $N_{p}$ = 18,800 (i.e., 0.05 photon/pixel). A photon-counting image with *N* = 1 observations is the same as a photon-counting image produced by conventional photon-counting imaging. As shown in Figure 11, the greater the number of observations the better the obtained visual quality was. To evaluate the visual quality of photon-counting images with different observations, we calculated performance metrics such as the peak sidelobe ratio (PSR), structural similarity (SSIM), the peak signal-to-noise ratio (PSNR), and peak-to-correlation energy (PCE), as shown in Figure 12. Among these performance metrics, PSR and PCE were calculated as follows:(20)$$c={\left|IFFT\left({\left(|I_{p}\left(\mu \right)||I_{ref}\left(\mu \right)|\right)}^{k}\times e^{j\phi_{I_{p}}-\phi_{I_{ref}}}\right)\right|}^{2}$$
(21)$$PSR=\frac{c_{max}-\bar{c}}{\sigma_{c}}$$
(22)$$PCE=\frac{{\left(c^{2}\right)}_{max}}{\sum c^{2}}$$
where *c* is the correlation value of the *k*th law nonlinear filter [39], $I_{p}\left(\mu \right)$ is the Fourier transform of the photon-counting image, $I_{ref}\left(\mu \right)$ is the Fourier transform of the reference image, *k* is the nonlinear coefficient of the *k*th law nonlinear filter, $\phi_{I_{p}}$ is the phase for the Fourier transform of the photon-counting image, and $\phi_{I_{ref}}$ is the phase for the Fourier transform of the reference image. In addition, $c_{max}$ is the maximum value of the correlation values, $\bar{c}$ is the mean of the correlation values, $\sigma_{c}$ is the standard deviation of the correlation values, ${\left(c^{2}\right)}_{max}$ is the maximum value of the squared correlation values, and $\sum c^{2}$ is the total energy of the correlation value.

In Figure 12a, the PSR values for a single observation and 100 observations were 13.3691 and 86.0931, respectively. Note that 100 observations had a PSR value that was 6.5 times better than that of a single observation. In Figure 12b, SSIM for a single observation and 100 observations were 0.1752 and 0.7436, respectively. The SSIM of 100 observations was almost 4 times better than that of a single observation. In Figure 12c, the PSNRs for a single and 100 observations were 16.1721 and 21.0083 [dB], respectively. Their difference was almost 5 [dB], which implied that 100 observations also performanced better than a single observation. Finally, in Figure 12d, PCE for a single and 100 observations were $2\times 10^{-4}$ and 0.0115, respectively. In this case, PCE for 100 observations was approximately 54 times better than that of a single observation. As a result, for a photon-counting image, *N*-observation photon-counting imaging can enhance the visual quality of an image under photon-starved conditions.

Now, let us consider 3D reconstructed images obtained by *N*-observation photon-counting integral imaging, which can be implemented by Equations (Equation 13)–(Equation 19). Figure 13 shows 3D reconstruction results for a red car at a reconstruction depth of 190mm with a different number of observations where $N_{p}$ = 18,800. As shown in Figure 13, the red car is in focus and it is remarkable that the higher the number of observations used the better the visual quality of the obtained 3D images.

Figure 14 shows 3D reconstruction results for a white car at a reconstruction depth of 230 mm with a different numbers of observations where $N_{p}$ = 18,800. It was noted that the higher the number of observations, the better the visual quality of the obtained 3D images. In addition, to show the ability of 3D reconstruction, we reconstructed 3D images at various reconstruction depths by Equations (Equation 4) and (Equation 19). The 3D reconstruction results are shown in Figure 15.

To evaluate the visual quality of our 3D reconstruction results, we calculated the peak-to-correlation energy that was mentioned earlier. Figure 16 shows the peak-to-correlation energy for our 3D reconstruction results at 190 mm and 230 mm via various numbers of observations where $N_{p}$ = 18,800. In Figure 16a, PCE for a single observation and 100 observations at 190 mm were 0.0036 and 0.0274, respectively, where the result for 100 observations was approximately 7.54 times better than that for a single observation. In Figure 16b, PCE for a single observation and 100 observations at 230 mm were 0.0043 and 0.0278, respectively. PCE of 100 observations was almost 6.4 times better than PCE of a single observation. Therefore, *N*-observation photon-counting integral imaging can enhance the visual quality of 3D images under severely photon-starved conditions.

## 4. Conclusions

In this paper, we proposed *N*-observation photon-counting integral imaging for 3D image visualization under severely photon-starved conditions. In conventional photon-counting integral imaging, the reconstructed 3D images may not be visualized sufficiently due to a lack of photons. On the other hand, since the number of samples can increase by using multiple observations (i.e., multiple photon-counting image generations), in our proposed method, the estimated scene may be more accurate than a single observation statistically. In addition, when the number of photons cannot increase, conventional photon-counting integral imaging may not visualize 3D images due to a lack of photons. In contrast, our proposed method can visualize 3D images under these conditions by increasing the number of observations, which is the novelty of our proposed method. Therefore, we believe that our method can be utilized by various applications under severely photon-starved conditions, such as in an unmanned vehicle under inclement weather conditions, a low-radiation medical device, defense, astronomy, and so on. However, our method has several drawbacks. It requires a lot of processing time because it implements multiple observations of photon-counting imaging to obtain the photon-counting image. In addition, the accuracy of the estimation results may be insufficient because our method uses a uniform distribution as the prior information of the original scene. This may be solved by using Bayesian approaches that have a specific statistical distribution, such as a Gamma distribution, as the prior information. In addition, processing time can be enhanced by only detecting photons from the objects and reducing the number of observations. We will investigate these issues in future work.

## Figures and Tables

**Figure 1 sensors-24-01731-f001:**
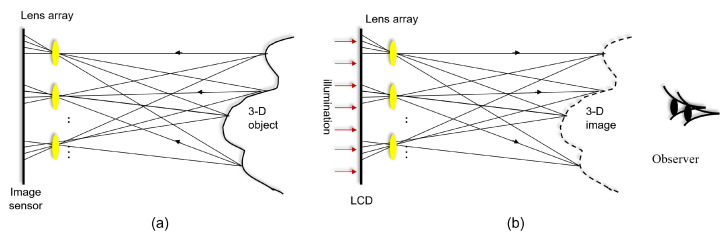
Integral imaging: (**a**) pickup and (**b**) display [26].

**Figure 2 sensors-24-01731-f002:**
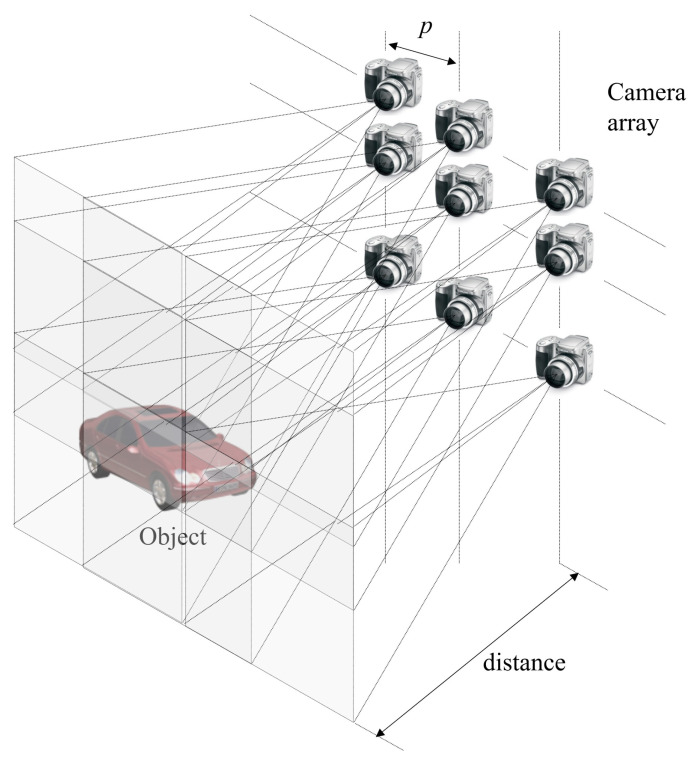
Synthetic aperture integral imaging (SAII).

**Figure 3 sensors-24-01731-f003:**
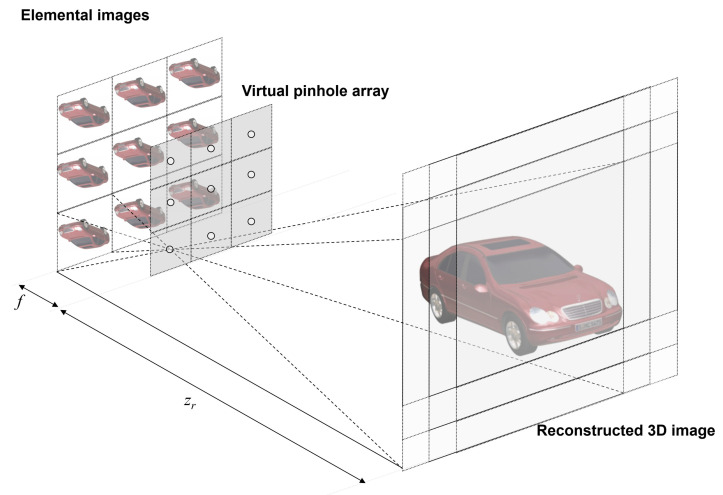
Volumetric computational reconstruction (VCR) of integral imaging [26].

**Figure 4 sensors-24-01731-f004:**
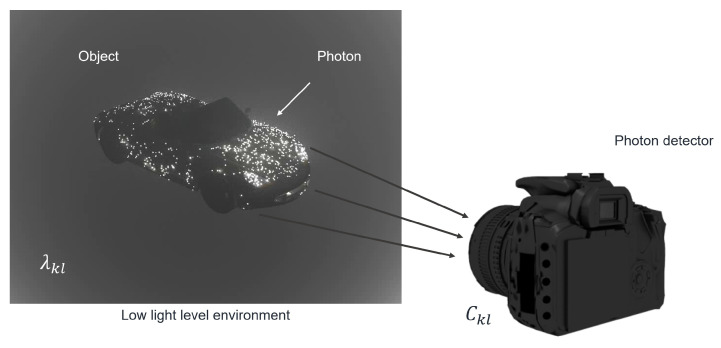
Physical photon-counting detector [37].

**Figure 5 sensors-24-01731-f005:**
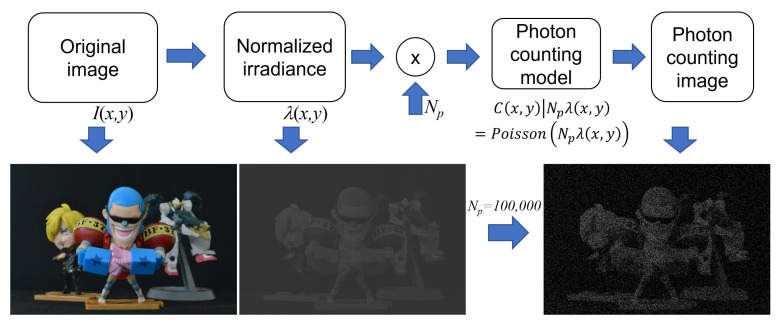
Computational photon-counting model.

**Figure 6 sensors-24-01731-f006:**
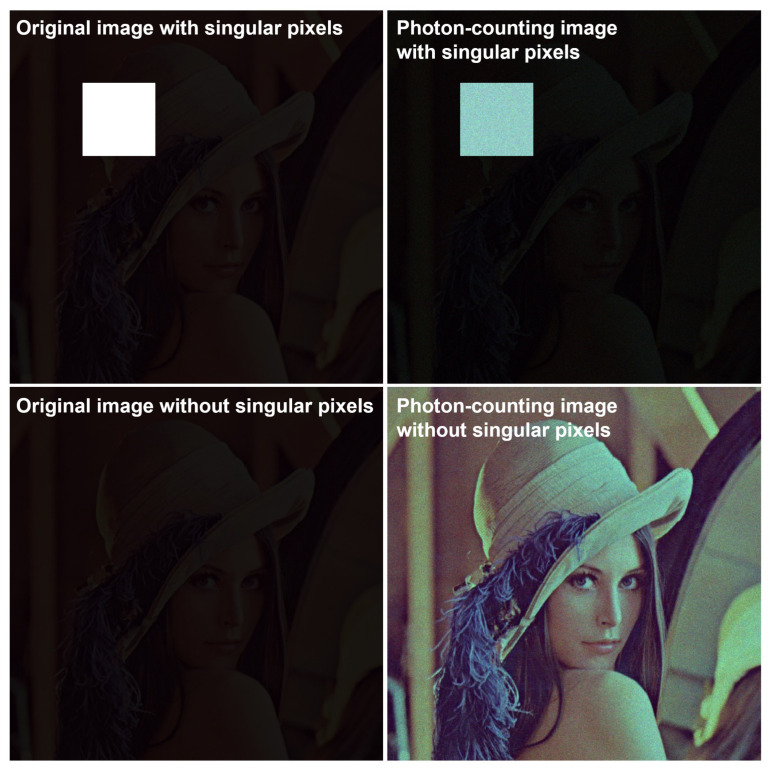
Photon-counting images with singular pixels.

**Figure 7 sensors-24-01731-f007:**
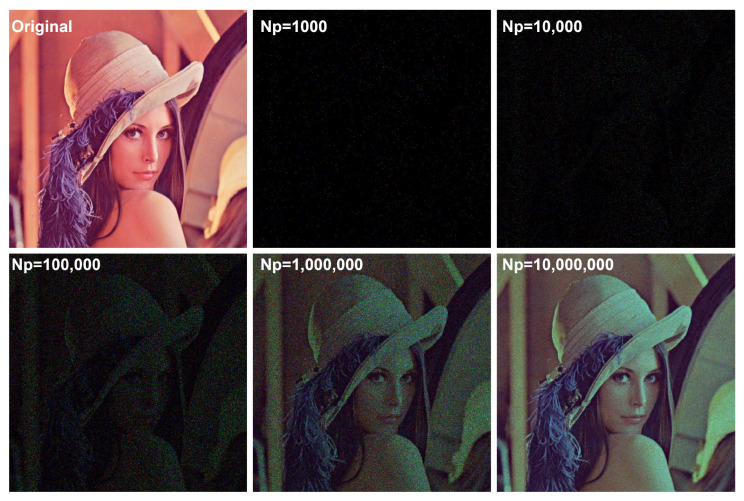
Photon-counting images with various number of photons.

**Figure 8 sensors-24-01731-f008:**
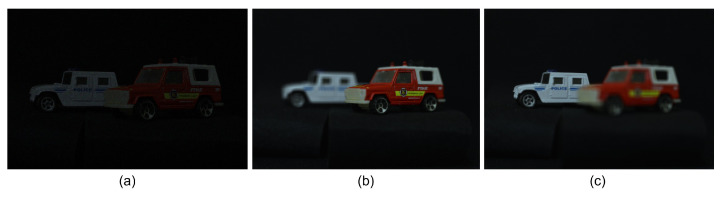
(**a**) Photon-counting image with $N_{p}$ = 1,500,000, (**b**) reconstructed 3D image by photon-counting integral imaging with MLE at $z_{r}$ = 190 mm, and (**c**) reconstructed 3D image by photon-counting integral imaging with MLE at $z_{r}$ = 230 mm.

**Figure 9 sensors-24-01731-f009:**
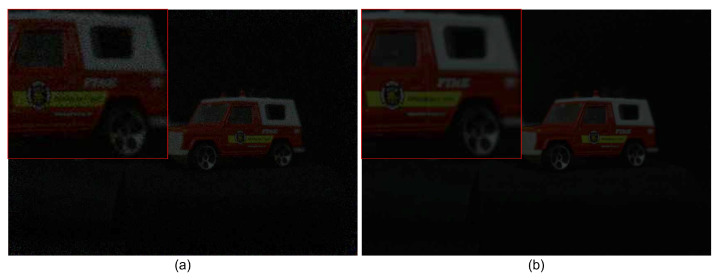
3D reconstructed images by (**a**) conventional photon-counting integral imaging and (**b**) our proposed method with 100 observations, where the expected number of photons is $N_{p}$ = 5000.

**Figure 10 sensors-24-01731-f010:**
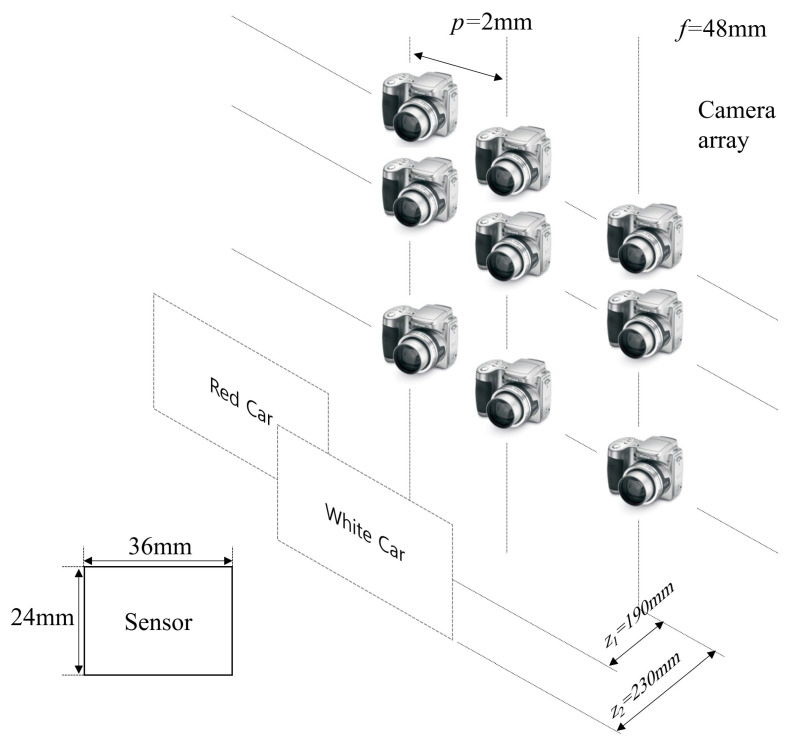
Experimental setup.

**Figure 11 sensors-24-01731-f011:**
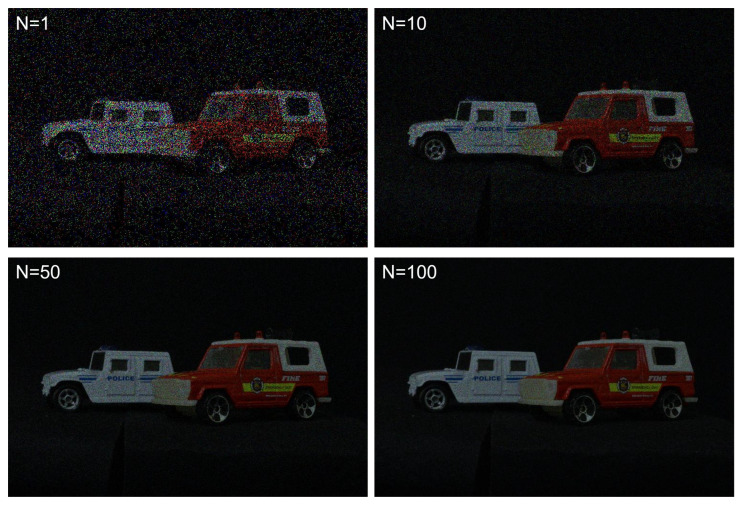
Photon-counting images with different observations where $N_{p}$ = 18,800.

**Figure 12 sensors-24-01731-f012:**
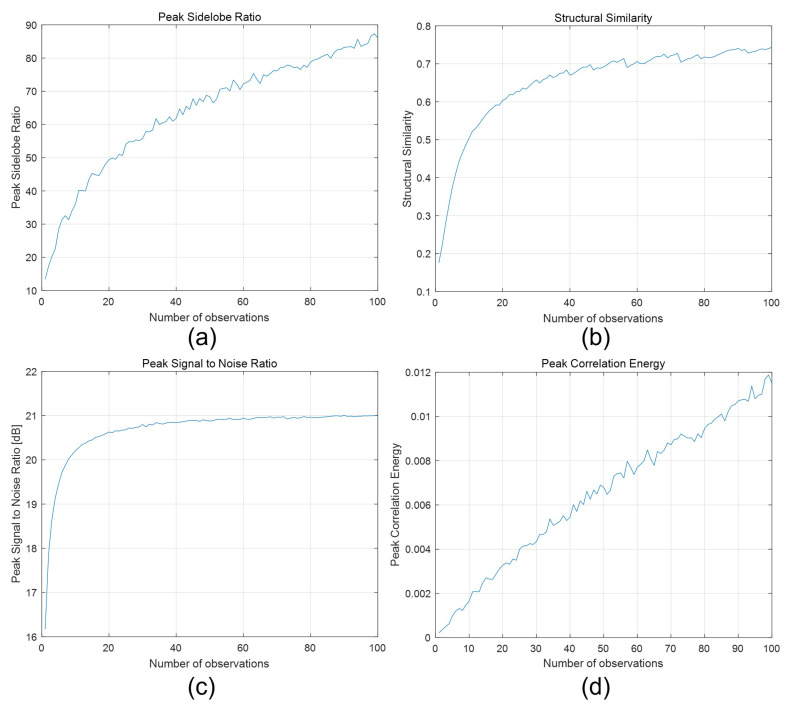
Performance metrics via different observations. (**a**) Peak sidelobe ratio, (**b**) structural similarity, (**c**) peak signal-to-noise ratio, and (**d**) peak-to-correlation energy, where the number of observations *N* ranged from 1 to 100.

**Figure 13 sensors-24-01731-f013:**
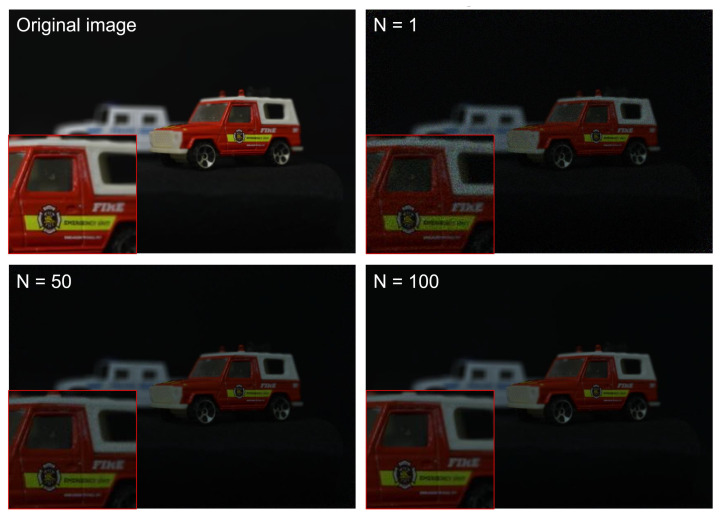
3D reconstruction results at a reconstruction depth of 190 mm with a different number of observations where $N_{p}$ = 18,800.

**Figure 14 sensors-24-01731-f014:**
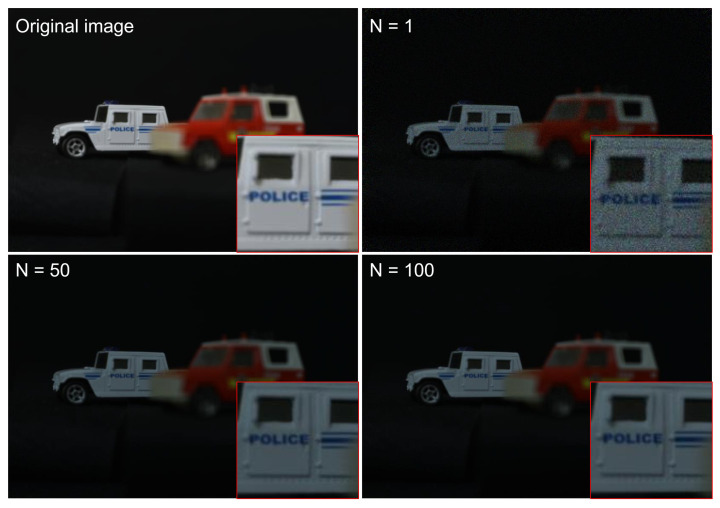
3D reconstruction results at a reconstruction depth of 230 mm with a different number of observations where $N_{p}$ = 18,800.

**Figure 15 sensors-24-01731-f015:**
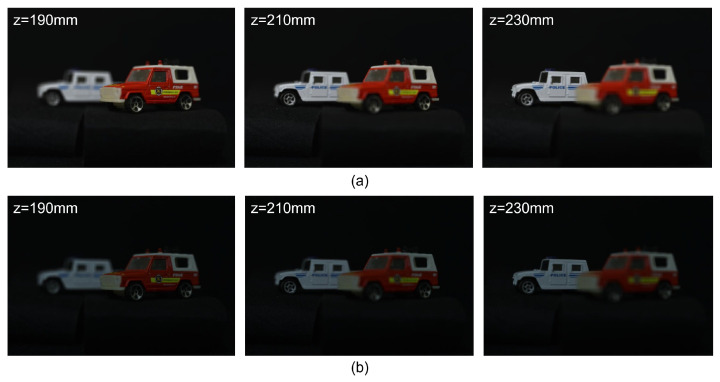
3D reconstruction results via different reconstruction depths using (**a**) original elemental images and (**b**) *N* = 20 observation photon-counting elemental images where $N_{p}$ = 18,800.

**Figure 16 sensors-24-01731-f016:**
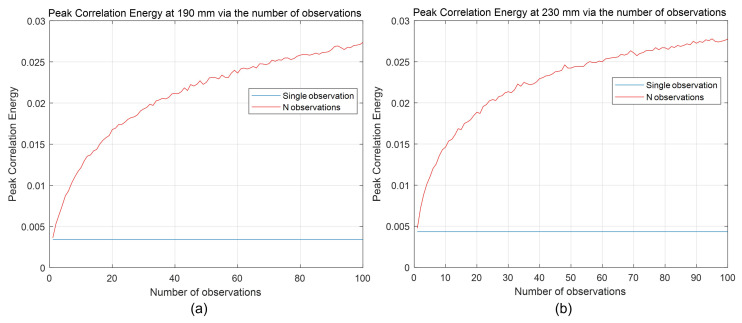
Peak-to-correlation energy for 3D reconstruction results (**a**) at 190 mm and (**b**) 230 mm via various number of observations where $N_{p}$ = 18,800.

## Data Availability

All data are contained within the article.

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
