# Peer review of "Three-Dimensional Image Visualization under Photon-Starved Conditions Using N Observations and Statistical Estimation"

_sensors, 2024, doi:10.3390/s24061731_

Round 1

Reviewer 1 Report

Comments and Suggestions for Authors

The article entitled "Three-dimensional Image Visualization under Photon-Starved Conditions using N Observation and Statistical Estimation" by M. Cho et al. presented a photon counting integral imaging (PCII) reconstruction via N observations. The article is well-written but has not proposed a significant contribution to the field of photon-counting imaging. Some of my major concerns are as follows:

1. The major claim of novelty is that the multiple observation-based PCII reconstruction. But it is a standard practice in PCII that, as the random (Poisson) process is involved, we either average or choose the best distribution.

2. Figure 6 depicts the impact of singular pixel distribution in photon counting systems. To alleviate this, a general practice is to choose a region of interest (ROI) and simulate the photon counting images. However, the authors did not explicitly refer to any technique to overcome the singular pixel issue.

3. In Figure 9, no visual difference is perceived when Np = 5000, even after 100 observations. However, in contrast, Figure 15 shows a significant improvement (visually) just with 20 observations when Np was set to 18,800. 

4. Finally, if the proposed approach can only demonstrate better visual quality at Np > 10(i.e., 18,800), it defeats the purpose of PCII. As can be seen from references 32–37 and other similar articles, typically PCII provides a better visual quality when Np = 103-104 without any additional computations or algorithms as proposed in the article. While I understand it completely depends on the scene, claiming an approach that only works on (typically) higher photon counts where most of the other scenes would already provide a better PSNR at that Np level nullifies the need for PCII.

For these reasons, I recommend rejecting this article.

Reviewer 2 Report

Comments and Suggestions for Authors

Authors have shown interesting study of photon counting integral imaging using multiple observations. However, I have several concerns in the manuscripts which need to be addressed before acceptance. 

1.     I have few naïve questions, like why so many cameras are used as array. Is there any dependence of array arrangement. 

2.     How sever is the photon starved, and how this is different with current photon starved 3D imaging methods. 

3.     I am curious, why lady image is used as example in fig 6,7. While, cars image are used in other figs?

4.     How different are figs 9 with figs 11. Fig 9 image should be similar with fig 11 (a), as fig 11(a) is also photon counting image with single observation like figs 9. 

5.     How come some cars image are focused and some not, like figs 11 are with both cars focused but figs 13 and 14 are with single car focused. 

6.     Please also mention at what distance image in fig 11 are taken. 

7.     Results in fig 15 are also bit confusing. Why images in 15(b) are worse than (a) even with more no of observations. Also, please explain in details about all the results in each figs.

8.     Finally, it looks like some relevant references need to be cited as following; 

a.     Sudheesh K. Rajput, Dhirendra Kumar, and Naveen K. Nishchal, "Photon counting imaging and phase mask multiplexing for multiple images authentication and digital hologram security," Appl. Opt. 54, 1657-1666 (2015) 

b.     Alok K Gupta and Naveen K Nishchal 2021 J. Opt. 23 025701

Comments on the Quality of English Language

line 73-74 looks unclear to me; 'To visualize 3D image under these conditions, photon-counting integral imaging 73 with statistical estimations.'

Reviewer 3 Report

Comments and Suggestions for Authors

Dear Editor,

In this manuscript, the authors have performed a 3D reconstruction of the images, and the obtained results are consistent with the title and abstract. Furthermore, the structure of the manuscript is also well organized and written S, the manuscript can be accepted after addressing the following comments.

1. As the authors wrote, the number of samples plays an important role in image reconstruction. What criteria do you use to select the minimum number of samples?

2. How is the processing speed compared to similar methods?

3. The importance of the proposed method is not well highlighted in the text.

4. What kind of detector was used? In what wavelength range is used in this research?

5. What is the processing power consumption?

6. What effect does ambient air quality have on the results? Are simulations designed for different conditions?

Kind regards,

Round 2

Reviewer 1 Report

Comments and Suggestions for Authors

Not Applicable. 

Comments on the Quality of English Language

The language usage is fine. 

Author Response

There is no comments.

Reviewer 2 Report

Comments and Suggestions for Authors

Thanks for answering my comments. I have few more comments;

1. I also want to know if imaging with different prospective is similar to imaging with multiple observations.  

2. It can also be interesting to compare with current 3D photon counting imaging, if there are any. 

Comments on the Quality of English Language

NA
